# The Role of Knee Flexors Hypertonia in the Decision-Making of Hamstring Lengthening Surgery for Individuals with Cerebral Palsy

Faustyna Manikowska [1],*, Sabina Brazevič [1], Marek Jóźwiak [1] and Maria K. Lebiedowska [2]

[1] Department of Pediatric Orthopedics and Traumatology, Poznań University of Medical Sciences, 61-701 Poznań, Poland

[2] Independent Researcher, Edmonton, AB T5R 2R2, Canada

* Correspondence: foka@interia.pl

**Abstract:** Popliteal angle (PA) and the knee position at the end of the swing phase during walking ($K_{ts}$) are considered criteria for treatment selection and outcome prediction of hamstring lengthening surgery in individuals with cerebral palsy (CP). However, the relationships among $K_{ts}$, PA, and hypertonia are not clear. This study aimed to determine whether hamstrings hypertonia affects the values of PA and $K_{ts}$, and how it may affect the hamstring lengthening decision for CP. One hundred and twenty-six subjects with CP (male = 72, female = 54; age = 11.1 ± 3.9 years) underwent gait analysis and examination of hamstrings hypertonia (Tardieu scale) and length (PA). We found that $K_{ts}$ increased with PA ($K_{ts}$ = 5.00 + 0.31 × PA (r = 0.39; $p < 0.001$)). Every 10° change of PA leads to 3° improvement of knee position in walking. $K_{ts}$ were larger ($p < 0.001$) in the limbs with (20.40 ± 11.27°) than without (15.60 ± 9.99°) knee flexors hypertonia (Tardieu slow); and were larger ($p < 0.001$) in the limbs with (20.39 ± 11.01°) than without (14.85 ± 9.89°) knee flexors hypertonia (Tardieu fast). PAs were larger ($p < 0.05$) in the limbs with (42.81 ± 12.66°) than without (38.96 ± 14.38°) hypertonia (Tardieu fast). $K_{ts}$ = 13.93° and PA = 30° were cutoff values of the presence of hypertonia with sensitivities of 75.0% and 89.1%, respectively. $K_{ts}$ increased with the PA in ambulatory CP. It is estimated that pathological increase of $K_{ts}$ occurs at PA ≥ 40°. The hypertonia of knee flexors affected $K_{ts}$ and PA. The presence of knee flexors hypertonia should be considered in the decision-making of hamstring lengthening for individuals with CP if $K_{ts}$ ≥ 13.93° and PA ≥ 30°.

**Keywords:** cerebral palsy; hypertonia; popliteal angle; gait analysis; hamstring lengthening

## 1. Introduction

Cerebral palsy (CP) is a group of neuromuscular disorders caused by early injury of the brain region involving movement pathways control. Neuromuscular deficits lead to functional limitations including abnormal gait patterns. Functional limitations are the consequences of complex impairments typical for upper motoneuron damage including weakness, lack of selective muscle control, and hypertonia due to spasticity and/or dystonia and/or muscle shortening.

The data from clinical tests and instrumental methods are used in daily clinical practice to determine an optimal treatment strategy. The popliteal angle (PA) test is commonly considered a passive assessment of the biomechanical properties of the knee flexors, in particular its length. During the standard PA procedure, the examiner extends the knee joint at a slow velocity while the examinee lies supine with the hip 90° flexed. The value of the PA may be related to increased resistance at the knee joint (Figure 1). The resistance may be increased due to biomechanical (muscle shortness) or neural factors (predominantly spastic hypertonia) as a result of velocity- or muscle-length-dependent stretch reflexes in patients with CP [1–3]. The most common tests for hypertonia include the modify

Ashworth scale (MAS) and the Tardieu scale (TAS), where TAS better complies with the concept of spasticity evoked by either slow or fast stretch [4].

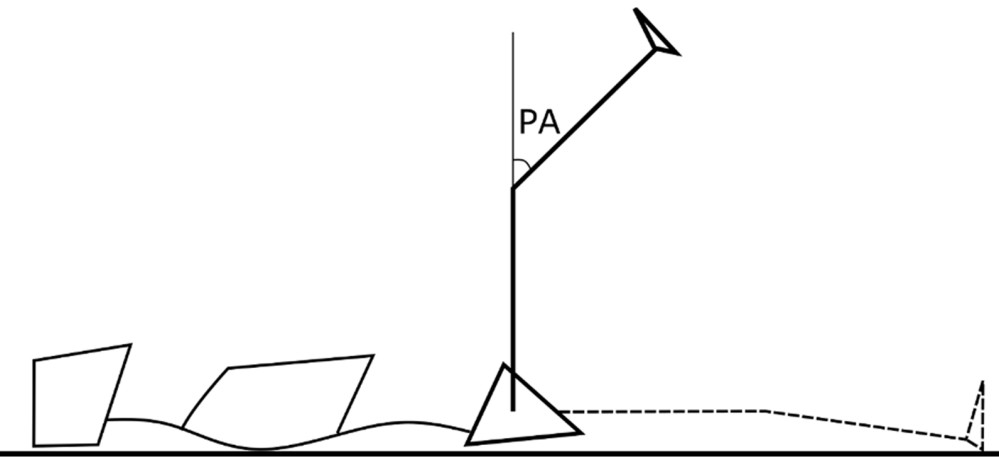

**Figure 1.** Standard popliteal angle (PA) test.

The instrumented three-dimensional gait analysis can quantify gait abnormality in individuals with neuromuscular deficits. Limited knee flexion range of motion (ROM) during gait is commonly observed in individuals with CP. The value of knee flexion at the end of the swing phase during walking ($K_{ts}$) is one of the criteria for treatment selection (Figure 2). Both muscle shortness and spastic hypertonia may affect the knee ROM during walking. The exaggerated stretch reflexes in response to muscle stretch could affect gait [5–9]. As a consequence, it is difficult to differentiate whether limitations in dynamic knee ROM during walking originate from hypertonia or shortness of knee flexors.

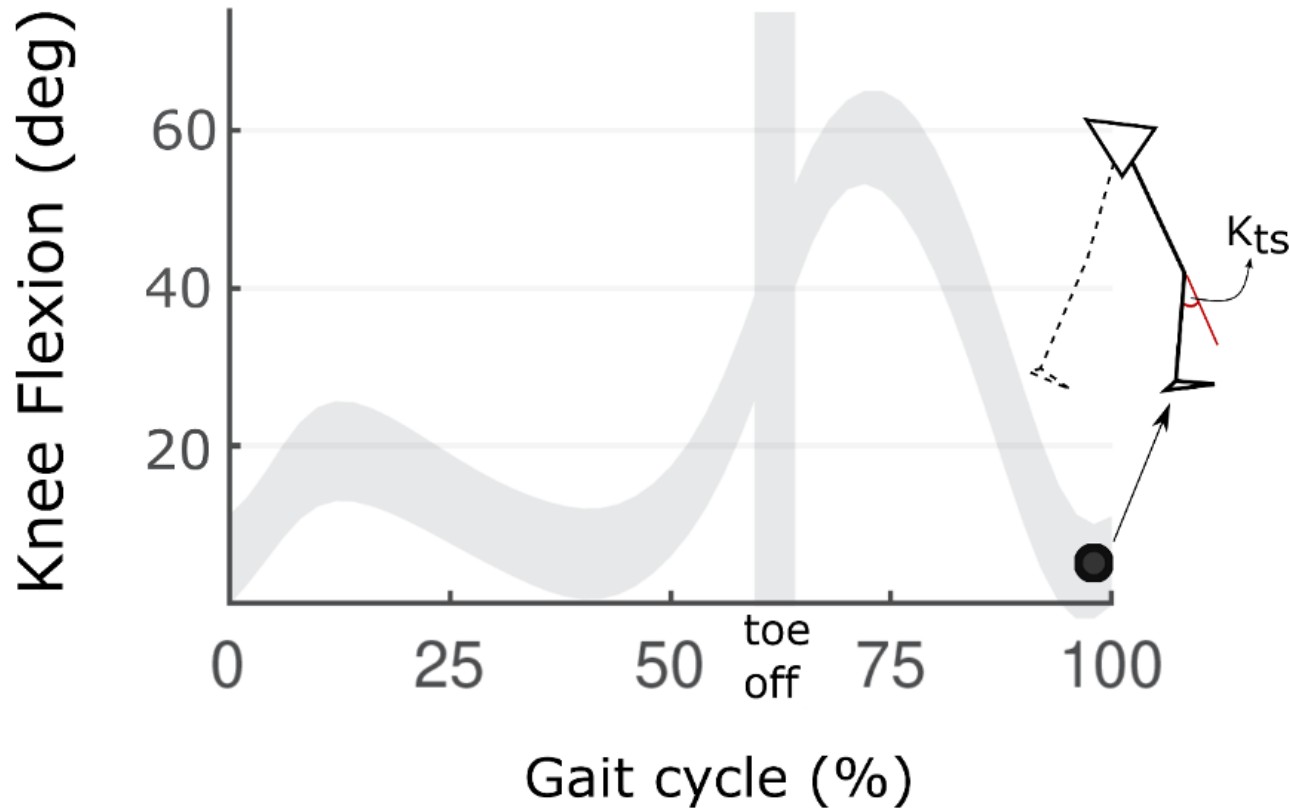

**Figure 2.** Knee flexion angle at the end of the swing phase during walking ($K_{ts}$).

The role of PA test as a criterion for hamstring lengthening surgery and as a predictor of knee flexion during walking is debatable because the conditions of testing on an examination bench are radically different than during walking [10]. Moreover, increasingly researchers emphasize the influence of hypertonia on PA test. The activation of ipsilateral and contralateral muscles of the legs has been reported [1,11,12]. The hamstring lengthening surgery is not always effective and the improvement of knee ROM during walking is not consistent across individuals [13–15]. Thus, muscle shortness may be only one of the contributing factors to gait pathology. Emerging data recognize improper links between neurological injury, neuromuscular deficits, and gait abnormalities in patients with CP. It seems that data obtained with gait analysis and clinical tests are not direct and obvious [16–18].

PA is a result of clinical evaluation performed on the examination table. However, the $K_{ts}$ is a knee position during walking which is a functional performance. Both values might be affected by hamstrings shortness and/or spastic hypertonia. It is unclear whether there exists an association between both of them. There is also no agreement on if different treatment procedures should be beneficial whether hypertonia is present or absent, especially since hypertonia may lead to, or coexist with, contractures. A better understanding of these relationships can lead to more effective treatment strategies.

The aims of the present study were:

1. To establish the association between PA and $K_{ts}$ in children with CP;
2. To determine whether hamstrings hypertonia affects the values of PA and $K_{ts}$;
3. To determine the probabilities of inclusion for hamstring lengthening surgery limbs of patients with or without hypertonia based on the values of PA and $K_{ts}$.

To achieve the above aims we analyzed the results of clinical tests: PA, muscle hypertonia (evaluated by TAS at both slow (i.e., Tardieu V1) and fast (i.e., Tardieu V3) velocities), and knee angular position at the end of swing (evaluated by instrumented gait analysis).

## 2. Materials and Methods

The appropriate Institutional Review Board approved the study (Approval No: 244/20 (11 March 2020), and written consent for using the examination data was acquired from all participants. For participants under the age of 18, consent was obtained from a parent or legal guardian.

### 2.1. Participants

In total, data from 126 individuals (male, n = 72; female, n = 54; age = 11.1 ± 3.9 years; 213 limbs) with spastic CP were retrospectively analyzed. All data were extracted from the database of the local gait analysis laboratory. Inclusion criteria were: (1) diagnosed as spastic CP; (2) no missing data in either clinical examinations or gait analysis. The exclusion criterion was any surgery or botulinum toxin within 6 months before the assessment.

### 2.2. Protocol

The selected gait parameters of patients were obtained through an eight-camera instrumented three-dimensional gait analysis system (Vicon, Vicon Motion Systems Ltd., Oxford, UK) sampling at 120 Hz. Fifteen reflective markers were attached to each participant based on the standard Plug-in-Gait model [19]. Data were collected only from the patients who were able to walk barefoot along the 10-m long examining walkway at least 3 times with or without the use of an assistive device. The $K_{ts}$ was measured for each gait cycle and the average values across the gait cycles were used in the analysis.

Selected clinical parameters were obtained through clinical evaluation of hamstrings muscle spasticity and measure of unilateral PA under standard positions [20,21]. Clinical examination was performed by experienced physical therapists with practice in the assessment of children with CP:

- Unilateral popliteal angle test (PA) was performed on the examination table with the participant in a supine position; the opposite leg relaxed on the table with both hip

and knee extended at the neutral position; the hip of the tested leg was flexed 90° and the examiner extended the knee. The angle was measured by a manual goniometer.

- Hamstrings muscle spasticity (HYP) was graded by TAS from 0 to 5 (where 0 means no resistance throughout the passive movement; 1 = slight resistance throughout passive movement without a catch; 2 = clear catch at a precise angle; 3 = fatigable clonus < 10 s; 4 = non-fatigable clonus > 10 s; 5 = joint immovable) in two velocities: slow (vs) and as fast as possible (vf). The spastic hypertonia was categorized into two groups depending on the presence (YES; i.e., TAS = 1–5) or absence (NO; i.e., TAS = 0) of impairment.

### 2.3. Outcome Measures

Primary outcome measures for the study were data from clinical tests: PA, TAS at slow velocity ($HYP_{vs}$), TAS at fast velocity ($HYP_{vf}$), and knee angular position at the end of swing phase ($K_{ts}$). Impairments of hypertonia (HYP) were categorized into two categories depending on the presence (YES) or the absence (NO) of it.

Secondary outcome measures were the cutoff values, i.e., the angular values of PA and $K_{ts}$ which separate the presence and absence of hypertonia (as evaluated with TAS), and angular values of PA which separate between presence and absence of lack of knee extension—$K_{ts} > 10°$ (taking into consideration the customarily accepted [22,23] 5° error of knee position measurement).

### 2.4. Data Analysis

Both PA and $K_{ts}$ were not normally distributed as confirmed with Kolmogorov–Smirnoff test and Spearman's correlation and the linear regression equations were calculated to determine the association between them. To determine whether hamstrings muscle hypertonia affects the values of PA and $K_{ts}$, Kruskal–Wallis ANOVA by ranks analysis was applied. To determine the probabilities of inclusion for hamstring lengthening surgery for patients with hamstrings shortening with or without hypertonia, we performed the sensitivity/specificity analysis of detection of hypertonia of hamstring muscles based on the spectrum of PA angle values (5.0–80.0°) and $K_{ts}$ values (0.6–52.6°).

The cutoff value for optimum sensitivity and specificity (to detect or not to detect hypertonia) of PA and $K_{ts}$ were calculated using receiver operating characteristic (ROC) analysis [24,25]. The strength of the impact was evaluated with the area under the sensitivity/specificity curve (AUC). The strength of the impact was evaluated with the area under the AUC where 1.0 means the strongest and 0.5 means no impact with a statistical significance of the *p*-value. Z-statistics were used to determine the statistical significance of the difference between the AUC and 0.5. The AUC scores ranged from 0.5 to 1.0: An AUC closer to 1.0 implied a stronger contribution of hypertonia on the PA/$K_{ts}$ value. The closer the AUC was to 0.5, the lower its contribution was. For the sake of the analysis, we defined the presence or absence of knee flexors hypertonia, which was based on the $HYP_{vs}$ and $HYP_{vf}$.

Statistical analysis was performed with Statistica13™, StatSoft™ Ltd., and Stata Cytel with a level of significance at 0.05.

## 3. Results

### 3.1. Association between PA and $K_{ts}$

We found that the knee flexion at the end of swing of walking ($K_{ts}$) increased with the popliteal angle (PA) according to the equation:

$$K_{ts} = 5.00 + 0.31 \times PA$$

r = 0.39; *p* < 0.001 in the limbs of ambulatory patients with CP (Figure 3).

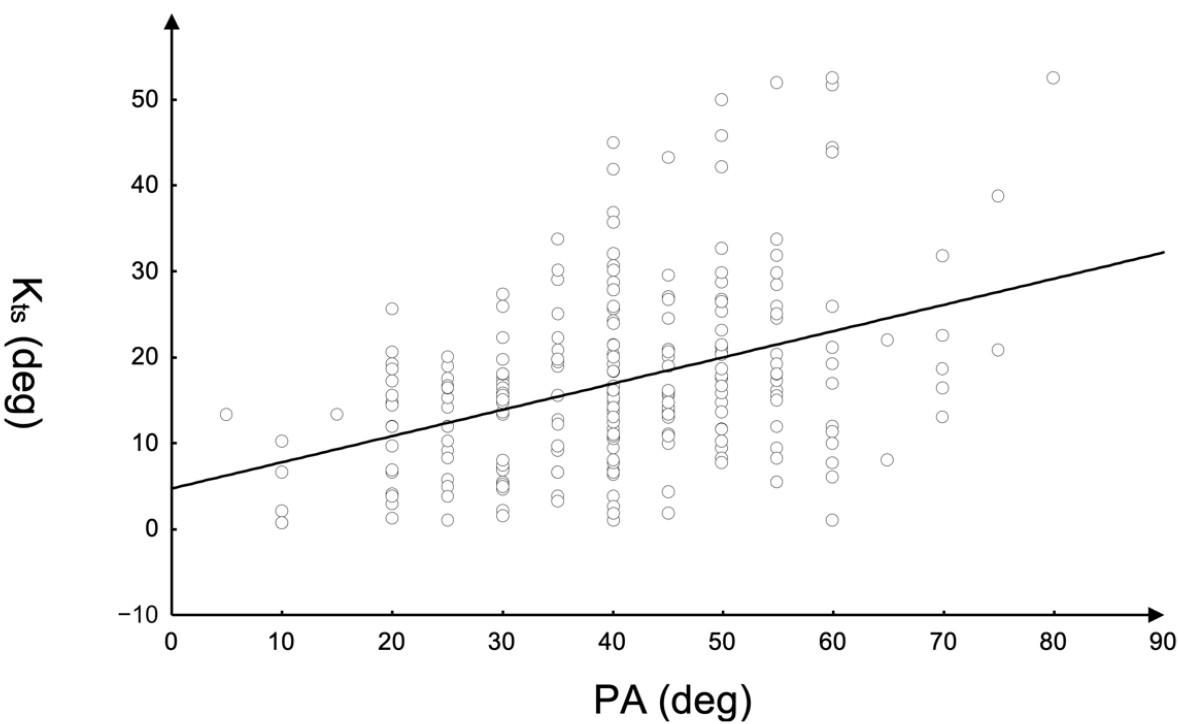

**Figure 3.** Knee position at the end of swing/initial stance ($K_{ts}$) increases with the popliteal angle (PA) according to the equation: $K_{ts} = 5.00 + 0.31 \times PA$ ($r = 0.39$; $p < 0.001$) in the limbs of patients with CP. Each circle represents the $K_{ts}$ value for the specific PA angle acquired from one single study participant.

The results of receiver-operating characteristic (ROC) analysis ($p = 0.05$, AUC = 0.634) indicate that the cutoff value of PA to detect 10° and more of knee flexion in $K_{ts}$ is 40° (sensitivity = 69.7%, specificity = 56.3%). For PA $\geq$ 40° the probability of the presence of pathological knee flexion in $K_{ts}$ increases.

### 3.2. Effect of HYP on PA and $K_{ts}$ in Children with CP

Our data indicate that the presence or absence of clinically recognized hamstrings hypertonia affected almost all variables. $K_{ts}$ was affected (Kruskal–Wallis test: H(1,N = 213) = 12.56, $p = 0.0004$)) by the presence of hypertonia (as recognized with $HYP_{vs}$) (Table 1).

**Table 1.** The popliteal angle values (PA) and knee angular position at the end of swing/initial stance ($K_{ts}$) were larger in the limbs with clinically detected knee flexors hypertonia, but the differences were not statistically significant for PA when hypertonia had been identified using TAS at slow velocities ($HYP_{vs}$).

| | Absence of Hypertonia (n = 117) | | | | | Presence of Hypertonia (n = 96) | | | | | *p* |
|---|---|---|---|---|---|---|---|---|---|---|---|
| Angle (deg) | Mean | Median | Min | Max | SD | Mean | Median | Min | Max | SD | |
| PA | 39.66 | 40.00 | 5.00 | 70.00 | 14.03 | 42.60 | 40.00 | 10.00 | 80.00 | 12.96 | 0.23 |
| $K_{ts}$ | 15.60 | 14.50 | 0.57 | 51.81 | 9.99 | 20.40 | 19.09 | 2.14 | 52.57 | 11.27 | **<0.001** |

n: number of limbs; SD: standard deviation; bold values denote statistical significance at the $p \leq 0.05$ level.

$K_{ts}$ and PA were affected by the presence of hypertonia (as recognized with $HYP_{vf}$) as examined with Kruskal–Wallis test: H(1,N = 213) = 16.26694, $p = 0.0001$ and H(1,N = 213) = 3.57, $p = 0.05$, respectively (Table 2).

**Table 2.** The popliteal angle values (PA) and knee angular position at the end of swing/initial stance ($K_{ts}$) were larger in the limbs with clinically detected knee flexors hypertonia, when hypertonia had been identified using TAS at fast velocities ($HYP_{vf}$).

| | Absence of Hypertonia (n = 101) | | | | | Presence of Hypertonia (n = 112) | | | | | *p* |
|---|---|---|---|---|---|---|---|---|---|---|---|
| Angle (deg) | Mean | Median | Min | Max | SD | Mean | Median | Min | Max | SD | |
| PA | 38.96 | 40.00 | 5.00 | 80.00 | 14.38 | 42.81 | 40.00 | 10.00 | 75.00 | 12.66 | **0.05** |
| $K_{ts}$ | 14.85 | 13.55 | 0.57 | 52.41 | 9.89 | 20.39 | 18.97 | 2.14 | 52.57 | 11.01 | **<0.001** |

n: number of limbs; SD: standard deviation; bold values denote statistical significance at the $p \leq 0.05$ level.

The PA and $K_{ts}$ were larger in the limbs with clinically detected knee flexors hypertonia, but the differences were not statistically significant for PA when hypertonia had been identified using TAS at slow velocities ($HYP_{vs}$). $K_{ts}$ were larger ($p < 0.001$) in the limbs with ($20.4 \pm 11.27°$) than without ($15.60 \pm 9.99°$) knee flexors hypertonia ($HYP_{vs}$). At fast velocity ($HYP_{vf}$), PAs were larger ($p \leq 0.05$) in the limbs with ($42.81 \pm 12.66°$) than without ($38.96 \pm 14.38°$) hypertonia and $K_{ts}$ were larger ($p < 0.001$) in the limbs of patients with ($20.39 \pm 11.01°$) than without ($14.85 \pm 9.89°$) knee flexors hypertonia.

*3.3. Probabilities of Inclusion for Hamstring Lengthening Surgery Patients with or without Hypertonia Based on the Values of PA and $K_{ts}$*

The analysis of sensitivity/specificity of $K_{ts}$ (Figure 4) to detect the presence of knee flexors hypertonia using ROC analysis (AUC = 0.66, $p < 0.0001$) (Table 3) indicates that $13.93°$ is the $K_{ts}$ cutoff value (sensitivity = 75%, specificity = 51.5%) to detect the absence of knee flexors hypertonia. The probability of absence of hypertonia (as detected with TAS at fast velocities $HYP_{vf}$) decreases for $K_{ts}$ larger than $13.93°$.

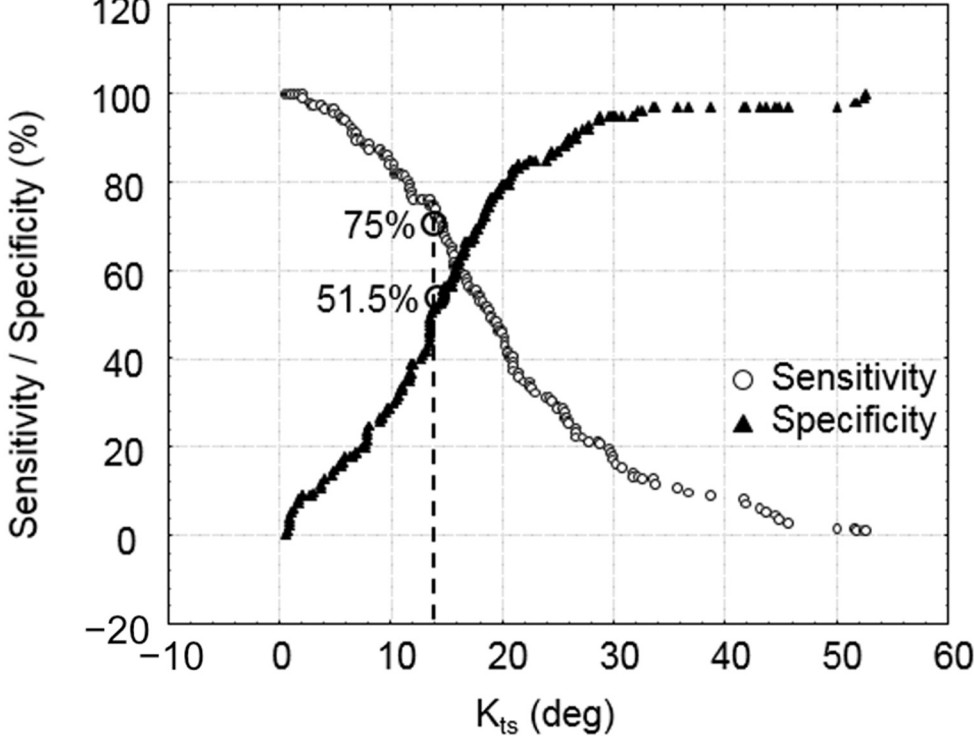

**Figure 4.** The results of ROC analysis (AUC = 0.66, $p < 0.0001$) indicate that the value of $K_{ts} = 13.93°$ generates the cutoff value (sensitivity = 75%, specificity = 51.5%) to detect the absence of knee flexors hypertonia. For $K_{ts} \geq 13.93°$, the probability of absence of hypertonia (as detected with TAS at fast velocities $HYP_{vf}$) decreases.

**Table 3.** The cutoff values of popliteal angle (PA) and knee angle at the end of swing/initial stance ($K_{ts}$) during gait to detect the presence of knee flexors hypertonia based on two clinical tests: TAS at slow ($HYP_{vs}$) and TAS fast ($HYP_{vf}$) velocities.

| | AUC | Cutoff (deg) | Sensitivity (%) | Specificity (%) | *p* |
|---|---|---|---|---|---|
| $K_{ts}$ ($HYP_{vs}$) | 0.641 | 16.68 | 61.46 | 64.96 | **0.000395** |
| $K_{ts}$ ($HYP_{vf}$) | 0.660 | 13.93 | 75.00 | 51.48 | **0.000055** |
| PA ($HYP_{vs}$) | 0.562 | 30.00 | 90.20 | 24.43 | 0.102084 |
| PA ($HYP_{vf}$) | 0.587 | 30.00 | 89.08 | 25.44 | **0.021889** |

The strength of the impact was evaluated with the area under the sensitivity/specificity curve (AUC) where 1 means the strongest and 0.5 no impact with a statistical significance of the *p*-value. Bold values denote statistical significance at the $p \leq 0.05$ level.

The analysis of sensitivity/specificity of PA (Figure 5) to detect the presence of knee flexors hypertonia using ROC analysis (AUC = 0.587, *p* = 0.022) (Table 3) indicates that 30° is the PA cutoff value (sensitivity = 89.07%, specificity = 25.44%) to detect the absence of knee flexors hypertonia. The probability of absence of hypertonia (as detected with TAS at fast velocities [$HYP_{vf}$]) decreases for PA larger than 30°.

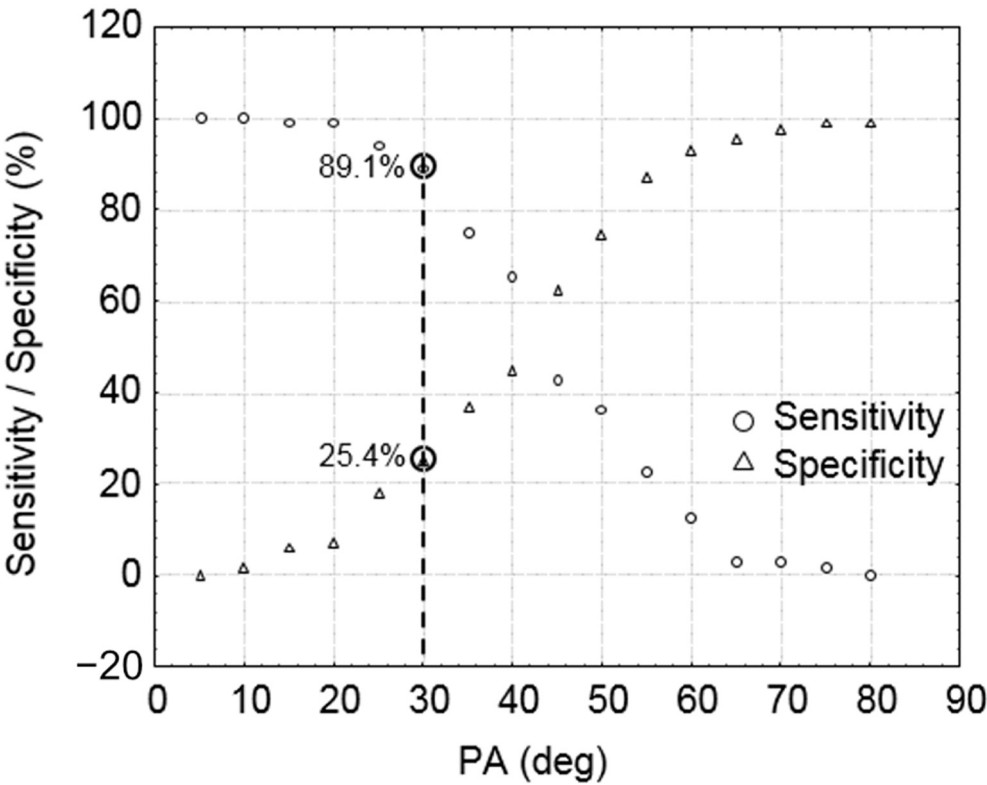

**Figure 5.** The results of ROC analysis (AUC = 0.587, *p* = 0.022) indicate that the value of popliteal angle (PA = 30°) generates the cutoff value (sensitivity = 89.07%, specificity = 25.44%) to detect the absence of knee flexors hypertonia. For PA ≥ 30°, the probability of absence of hypertonia decreases.

ROC analysis suggested that $K_{ts} \geq 13.93°$ (sensitivity = 75.0%) and PA ≥ 30° (sensitivity = 89.1%) were affected by the presence of hypertonia.

## 4. Discussion

The first aim of the present study was to establish the association between popliteal angle (PA) and knee angle at the end of the swing phase in walking ($K_{ts}$) in children with CP. We found a weak correlation between clinically obtained values of PA and $K_{ts}$ that suggests an association between them.

The differences between the conditions in which the clinical and instrumental data had been collected could be responsible for 15% of the variance of $K_{ts}$ due to PA. Taking into consideration that an increase in PA is related to an increase in $K_{ts}$ according to the equation $K_{ts} = 5.00 + 0.31 \times PA$ ($r = 0.39$; $p < 0.001$) it can be estimated that a decrease in PA by 10° can lead to 3° improvement of knee position in walking. The value of the estimation can be tested using the data from pre- and post-surgery if available.

We found that the presence or absence of hamstrings hypertonia (clinically detected with TAS at slow and fast velocities) affected the majority of variables (Tables 1 and 2). This suggests the contribution of hamstrings hypertonia to the increased values of PA and to an increased knee flexion at the end of swing (Aim 2). The values of PA and knee position at the end of swing were larger (from 11% to 39% when means were considered) in limbs with, rather than without, hypertonia. It had been previously acknowledged that the lack of knee extension at the end of swing may be related not only to hamstrings muscle shortness [26,27], but also to hamstrings muscle hypertonia and/or selective motor control impairment in a patient with CP [27–31]. Modelling data also suggest that other factors beyond hamstrings muscle shortness seem to affect knee position at the end of swing [32,33].

The associations between hamstrings spasticity, PA, and knee position at the end of swing remain debatable [34–37]. Both the presence [34–36] and absence [4,37] of such relationships have been previously reported. Muscle activation, also triggered by stretch reflexes leading to an increased resistance (hypertonia), is commonly reported in patients with CP [38].

It has been reported that PAs were larger when hamstrings were activated than when they were silent in the limbs of patients with CP [1,11,12,39].

Our data provide another argument to support the impact of hamstrings muscle spastic hypertonia on the increased PA values and an increased knee flexion at the end of swing phase of walking. The strength of the impact of each of the factors requires further studies to better customize the treatment to individual patients' needs.

The application of ROC analysis allows the determination of the probabilities of inclusion for hamstring lengthening surgery for patients with or without hypertonia based on the cutoff values of PA and $K_{ts}$. The results of ROC analysis not only conforms with another method that had been used in a previous aims to show that both $K_{ts}$ and PA can be used to differentiate patients' limbs with the presence or absence of clinically tested hypertonia (Table 3), but also allows us to establish the cutoff values for such differentiation.

We found that $K_{ts} = 13.93°$ (AUC = 0.66, $p < 0.0001$) generates the cutoff value (sensitivity = 75%, specificity = 51.5%) to detect the absence of knee flexors hypertonia. For $K_{ts} \geq 13.93°$ the probability of absence of hypertonia decreases rapidly (as detected with TAS at fast velocities) (Figure 4). The value of PA = 30° (AUC = 0.587, $p = 0.022$) generates the cutoff value (sensitivity = 89.07%, specificity = 25.44%) to detect the absence of knee flexors hypertonia. The probability of absence of hypertonia decreases for PA $\geq$ 30° (Figure 5). In addition, we found that the pathologically decreased knee extension at the end of swing $K_{ts} \geq 10°$ [22,23] can be predicted with 69.7% of sensitivity and 56.3% of specificity when PA = 40° with rapidly increasing sensitivity for PA > 40°.

To our knowledge, our study provides original, analytically obtained cutoff values of PA and $K_{ts}$ during gait to detect the presence of knee flexors hypertonia based on two clinical tests: TAS at slow and TAS at fast velocities ($HYP_{vs}$, $HYP_{vf}$, respectively) (Table 3). The value of such analysis is supported by an effort to establish the criteria for knee flexors lengthening surgery based on the values of clinically tested PA and $K_{ts}$, obtained during gait analysis, especially since it has been reported that they are affected as results of surgeries [5,11,12,15,40]. However, the improvement of knee position at the end of swing after hamstring lengthening was not always successful despite the fact of a decreasing value of PA [5,11,12,14,15,31,40].

It has been recently suggested that indications for hamstring lengthening surgery should focus on instrumented gait analysis data supplemented only by the results of clinical

tests in ambulatory children with CP [41]. It has been advocated that the lack of extension especially $K_{ts} > 30°$ at the end of swing/initial stance should be used as a main criterion for the hamstring lengthening surgery. Our data do not question this criterion; however, we would like to emphasize that in 98% of ambulatory individuals with CP, $K_{ts} \geq 30°$ indicates the coexistence of hypertonia (Figure 4). It seems that the interaction between muscle shortness and hypertonia should be considered before hamstring lengthening surgery [37] because the presence of hypertonia might influence the selection of appropriate treatment [42].

Theoretically, a hamstring lengthening surgery in the presence of muscle shortening and the absence of hypertonia should increase the knee extension range of the motion and result in both decrease in PA and knee flexion at the end of swing. Moreover, as a result of hamstring lengthening surgery the length of the muscle at certain positions is shorter, the knee flexors moment of force is smaller [43,44] and the stretch reflex activation also decreases with decreased muscle length as was found in human spasticity by Burke et al. in 1971. Thus, in both cases, the so-called muscle lengthening surgery could be beneficial to the normalization of the value of PA and functional improvement. Another important aspect of the interaction between contractures and hypertonia which requires further investigation is the potential effect of the type of hypertonia (velocity or position dependent), especially when a positive functional outcome of treatment, e.g., botulinum toxin injection, has been reported in the velocity- but not in the position-dependent hypertonia [10]. Future studies should reveal if different types of hamstring lengthening surgery (tendon vs. muscle) increase the muscle–tendon unit length, through a decrease (tendon) or increase (muscle) of muscle length that differently affect position- vs. velocity- dependent hypertonia.

In everyday clinical practice, not only the procedures but also arbitrary selection based on the experience and knowledge of individual orthopedic surgeons should also be accepted. Based on our data (Figures 3 and 4), the probabilities of inclusion of patients with hypertonia based on the other arbitrary selected values of PA and $K_{ts}$ can be estimated for the first time.

*Limitations*

The present study focused on hypertonia only. Other factors and impairments may also contribute to gait abnormalities in patients with CP [45]. Impacts from other impairments should be addressed in future studies. Moreover, in this study, we considered all participants with CP as one group. Dividing them into different groups based on their Gross Motor Function Classification System (GMFCS) levels or severity of symptoms could complement our findings [37]. In addition, we defined the presence or absence of hypertonia based on the results of clinical TAS at slow and fast velocities. An instrumental evaluation of hypertonia should provide more precise and quantitative measures which should be used in future studies. Lastly, we did not investigate structural properties related to muscle shortness.

## 5. Conclusions

$K_{ts}$ increased with the PA in ambulatory individuals with CP. It is estimated that pathological increase of $K_{ts}$ occurs at PA $\geq 40°$. The hypertonia of knee flexors affected $K_{ts}$ and PA. The presence of knee flexors hypertonia should be considered if $K_{ts} \geq 13.93°$ and PA $\geq 30°$, i.e., the cutoff values of the presence of hypertonia.

**Author Contributions:** Conceptualization, F.M. and M.K.L.; methodology, F.M. and M.K.L.; software, S.B.; validation, F.M., S.B., M.J. and M.K.L.; formal analysis, F.M., S.B. and M.K.L.; investigation, F.M. and S.B.; resources, M.J.; data curation, F.M., S.B., M.J. and M.K.L.; writing—original draft preparation, F.M. and M.K.L.; writing—review and editing, F.M., S.B., M.J. and M.K.L.; supervision, M.J. and M.K.L.; project administration, F.M. All authors have read and agreed to the published version of the manuscript.

**Funding:** This research received no external funding.

**Institutional Review Board Statement:** The study was conducted in accordance with the Declaration of Helsinki and approved by the Research Ethics Committees of Poznań University of Medical Sciences, Poland (approval #244/20 11 March 2020).

**Informed Consent Statement:** Informed consent was obtained from all subjects involved in the study. Written informed consent has been obtained from the patients to publish this paper.

**Data Availability Statement:** The datasets generated during and/or analyzed during the current study are available from the corresponding author on reasonable request.

**Acknowledgments:** The authors wish to thank Brian Po-Jung Chen for his perceptive editing of the paper.

**Conflicts of Interest:** The authors declare no conflict of interest.

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
