# Peer review of "The Role of Knee Flexors Hypertonia in the Decision-Making of Hamstring Lengthening Surgery for Individuals with Cerebral Palsy"

_applsci, doi:10.3390/app12189210_

Round 1

Reviewer 1 Report

In this paper, to determine the probabilities of inclusion for knee flexors (hamstring) lengthening surgery for children with cerebral palsy, the effect of hamstrings hypertonia on popliteal angle (PA) and knee angle position at the end of swing phase (Kts) is investigated. As a result, it is clarified that the hamstring hypertonia increases the values of PA and Kts, and the cutoff values of PA and Kts which can detect the presence of the hamstring hypertonia are obtained. Then, it is concluded that the presence of hamstring hypertonia should be considered in the decision making of the hamstring lengthening surgery if PA and Kts are above the cutoff values.

This paper seems to be practically useful and interesting. However, there exist some insufficient descriptions, as follows, that should be added and/or corrected.

(1) The definitions of popliteal angle (PA) and knee angle position at the end of swing phase (Kts) are difficult to understand, especially for non-specialists. They should be shown by diagrams.

(2) As mentioned in Conclusion, it is described that the presence of hamstring hypertonia should be considered in the planning of the hamstring lengthening surgery if PA and Kts are above the cutoff values, but it is a little ambiguous. Does it mean that the hamstring lengthening surgery should not be done in this case?

(3) In Abstract (line 1617), it is described “Every 10°change of PA leads to 8°improvement of knee position in walking”, and in Discussion (line 232233), it is also described “the decrease in PA by 10°can lead to 8 improvement of knee position in walking”. In both cases, 8°should be 3°.

(4) About Figure 2andFigure 3

In Figure 2,

Kneeterm Kts

The label of longitudinal axis “Sensitivity, Specificity” is missing.

66 75

In Figure 3,

The label of longitudinal axis “Sensitivity, Specificity” is missing.

(5) Correct the following careless mistakes.

Line 9 : Popliteal angle test (PA) Popliteal angle (PA)

Line17,18 : with (15.00±9.56°) than without (20.14±11.09°)

             → with (20.14±11.09°) than without (15.00±9.56°)

 Line 21 : Kts>13.93°and PA>30°were cutoff values

Kts=13.93°and PA=30°were cutoff values

Line 151, 165, 177, 238, 239, 244, 245, 277 and 283 :

 at the of swing at the end of swing

Author Response

The response PDF file has been attached.

Reviewer 2 Report

Dear Dr Manikowska and colleagues,

Thank you for your interesting work on the biomechanics of the lower limb in patients with cerebral palsy. I recommend your paper is accepted, after some minor revisions. My suggestions/questions are as follows.

  1. Section 1. Can you please add a diagram explaining the variables Kst and PA if possible? I think it would greatly assist the reader, especially when they are reviewing your results in the later section.

  2. Section 2.1. Can you please comment further on ethics considerations such as an approval number and informed consent? I noticed you mentioned retrospective; perhaps comment on these matters in relation to use of retrospective data.

  3. Section 2.2. Plug-in-Gait model. Can you please explain this in a sentence or 2, and/or with a reference if necessary? This may help a general informed reader who may not understand all the ins and outs of biomechanics research.

  4. Figure 1. Can you explain the data a little further? Each dot is data for each patient, at each Kst/PA? One sentence explaining this in the caption could help the reader.

  5. Table 3. Can you please add the threshold for statistical significance in the caption?

  6. Overall the paper is well-written, maybe just review one more time for any minor proofreading required.

All the best in preparing your final version!

Kind regards,

Author Response

(The authors gave the same response as above.)

Round 2

Reviewer 1 Report

In this paper, to determine the probabilities of inclusion for knee flexors (hamstring) lengthening surgery for children with cerebral palsy, the effect of hamstrings hypertonia on popliteal angle (PA) and knee angle position at the end of swing phase (Kts) is investigated. As a result, the cutoff values of PA and Kts which can detect the presence of the hamstring hypertonia are clarified. Then, it is concluded that the presence of hamstring hypertonia should be considered in the decision making of the hamstring lengthening surgery if PA and Kts are above the cutoff values.

This paper seems to be practically useful and interesting. It becomes almost acceptable by the previous review. However, there remain some insufficient descriptions in Fig. 2, as follows, that should be corrected.

About Figure 2

The definition of Knee angle position at the end of swing phase (Kts) in Fig. 2 is mistaken.

Since the title of Fig. 2 is wrong and the label (explanation) of horizontal axis is missing, Fig. 2 is very difficult to understand. The title should be corrected and the label of horizontal axis should be added.

The label of vertical axis : Knee Flexion Knee Flexion (Kts)

Author Response

Author's notes have be provided via the attached PDF file. Thank you.
